# Codon Usage Bias in Human RNA Viruses and Its Impact on Viral Translation, Fitness, and Evolution

**DOI:** 10.3390/v17091218

**Published:** 2025-09-06

**Authors:** Iván Ventoso

**Affiliations:** Centro de Biología Molecular Severo Ochoa (CSIC-UAM) and Departamento de Biología Molecular, Universidad Autónoma de Madrid, Cantoblanco, 28049 Madrid, Spain; iventoso@cbm.csic.es

**Keywords:** human RNA virus, codon usage bias, synonymous codons, tRNAs, translation, virus fitness, virus evolution

## Abstract

Synonymous codon usage (codon bias) greatly influences not only translation but also mRNA stability. In vertebrates, highly expressed genes preferentially use codons with an optimal tRNA adaptation index (tAI) that mostly end in C or G. Surprisingly, the codon usage of viruses infecting humans often deviates from optimality, showing an enrichment in A/U-ending codons, which are generally associated with slow decoding and reduced mRNA stability. This observation is particularly evident in RNA viruses causing respiratory illnesses in humans. This review analyzes the mutational and selective forces that shape nucleotide composition and codon usage drift in human RNA viruses, as well as their impact on translation, viral fitness, and evolution. It also describes how some viruses overcome suboptimal codon usage to outcompete host mRNA for translation. Finally, the roles of viral tropism and host adaptation in codon usage bias of prototypical viruses are discussed.

## 1. Introduction

Synonymous codons are decoded by isoacceptor tRNAs carrying the same amino acid but different anticodons. However, not all synonymous codons are used with the same frequency, depending on the organism and gene analyzed [1,2]. The resulting codon usage bias (CUB) can strongly influence the expression of protein-coding genes at both transcriptional and post-transcriptional levels, affecting splicing, translation, mRNA stability, or even mRNA localization [2,3,4]. During translation, mRNA codons are read by ribosomes following the rules originally described by Francis Crick [5,6]; the abundance of cognate isoacceptor tRNAs being what ultimately determines the rate at which a particular codon is decoded. Thus, the tRNA adaptation index (tAI) of codons can be used to classify codons as optimal or non-optimal in a particular species, tissue, or cell type, although other metrics, such as the codon adaptation index (CAI) or the effective number of codons, are still used [7,8]. Highly expressed mRNAs from housekeeping genes are rich in optimal “abundant” codons that correlate with high translation rates. In contrast, the presence of suboptimal “rare” codons can slow down translation elongation, resulting in ribosome stalling and/or collision, which can eventually trigger mRNA degradation pathways [2,9]. Thus, codon usage, translation, and mRNA stability are closely linked through pathways that sense and connect ribosomal flux with the main mRNA degradation activities of the cell [9,10]. The preferential use of certain synonymous codons over others (RSCU, relative synonymous codon usage) can vary among groups of organisms. Thus, the mean RSCU in mammals differs from that observed in arthropods, yeast, or bacteria, especially in certain codons, such as those coding for Arginine (Arg) [11]. Hence, efficient expression of heterologous genes often requires optimization to harmonize their codon usage with that of the host species [12]. This paradigm has historically influenced the way to interpret codon composition in eukaryotic viruses as a simple adaptation to the host tRNA supply, an idea that is now changing [13,14].

## 2. Codon Usage in RNA Virus

Since viral genomes are subjected to unique mutational and selective pressures, the codon usage and nucleotide/dinucleotide composition of viral genomes can deviate from that of their hosts [15]. Contrary to the codon usage harmonization observed in phage-bacterial host pairs, the codon usage of most human viruses does not match that of their hosts, especially when compared with the set of most abundant cellular mRNAs rich in optimal codons [14,16]. In general, each family of viruses has a distinctive nucleotide composition and codon usage; however, some members can deviate from this trend, showing a more biased codon usage, as is the case for hepatitis C [16,17,18]. CUB also depends on whether the viral genome is RNA or DNA, whether the replicative cycle is nuclear or cytoplasmic, or whether the virus alternates replication in evolutionarily distant hosts. Since CUB can also influence nuclear DNA transcription, DNA viruses have been intentionally omitted from this review. Some differences in codon usage were also found when comparing groups of viruses with (+) ssRNA, (−) ssRNA, and dsRNA genomes, although belonging to a virus family remains more defining [16,18]. The RSCU can also vary across the genome, such that some differences in CUB can be found among viral genes (see below). In general, RNA viruses exhibit a reduction in C/G-ending codons, which reaches a maximum in some human respiratory viruses, including SARS-CoV-2 and human respiratory syncytial virus (HRSV) (Figure 1A). The resulting enrichment of A/U-ending codons in these viruses could be interpreted as detrimental for virus fitness since they are associated with reduced translation and stability in many human mRNAs [2]. However, experimental evidence showed that these viral mRNAs can be efficiently translated in infected cells by outcompeting host mRNAs for ribosomes [19,20,21]. Most RNA viruses infecting humans showed a suboptimal genomic tAI (tAIg) value, in particular retrovirus (HIV-1) and respiratory virus (HRSV-A, HPIV-2), whereas those families of viruses that alternate replication in evolutionarily distant hosts (insects/mammals), such as Flavivirus and Togavirus, exhibited more optimal tAIg values and a higher number of C/G-ending (optimal) codons (Figure 1A).

Regarding the differential use of synonymous codons, a strong preference for AGA (Arg) and a drastic reduction in the use of CUG (Leu) are the most distinctive signatures of human RNA viruses compared with human tissues (Figure 1B). This agrees with a previous analysis indicating the differential use of Arg codons as the major contributor to differences in CUB across domains of life [11]. More specifically, viruses infecting the respiratory tract showed an even stronger positive bias for AGA (Arg) and negative selection for CUG (Leu), along with a significant increase in the use of UUA (Leu), among other differences (Figure 1B). Remarkably, the twenty-one respiratory viruses infecting humans showed a coherent codon bias despite their diversity, including members of up to four unrelated families (Coronaviridae, Picornaviridae, Paramyxoviridae, and Orthomyxoviridae). This suggests that these viruses have been subjected to similar mutational and/or selective pressures from replicating in the respiratory tract.

## 3. Forces That Shape Codon Usage in RNA Viruses

Many forces, including mutation, selection, and genetic drift, can shape the codon usage of viruses (Figure 2). Since viruses are subjected to intense purifying selection, mutations preferentially accumulate at the third position in viral codon sequences, without affecting the amino acid sequence (silent changes), but promoting codon bias in many cases [15,23,24]. Apart from the stochastic errors introduced by error-prone RNA virus replicases during replication, extensive analysis of viral sequences and functional assays have unveiled the role of two families of host nucleic acid editing enzymes in viral genome mutagenesis; the APOBEC3 (apolipoprotein-B mRNA-editing complex 3) family that deaminates cytidines (C) to uridines (U) on ssDNA and RNA, and the ADAR-1 (adenosine Deaminase Acting on ribonucleic acid 1) family that deaminates adenosine (A) to inosine (I) on dsRNA, thus inducing A to G transitions [25]. These two families of deaminases have undergone a notable genetic expansion in mammals, and their expression is induced by interferon (IFN), a finding that emphasizes their role in antiviral response. The deaminase activity of some members of APOBEC3 family on viral genomes has been demonstrated in HIV-1 and other retroviruses, coronavirus, HBV, as well as in some DNA viruses [25,26,27]. Interestingly, A3A and A3G members of APOBEC3 family also showed activity on RNA molecules, suggesting a role of these deaminases against RNA viruses [26,28]. Specifically, an increased C-to-U mutation accumulated in the SARS-CoV-2 genome during infection of human cells over-expressing APOBECA3A, a finding that mimics the “rampant” C-to-U mutagenesis observed during the evolution of SARS-CoV-2 during the pandemic and post-pandemic period (2020–2024) [29,30]. This strongly suggests a direct role of APOBECA3A in SARS-CoV-2 evolution. Since APOBEC3 deaminases show sequence context preferences (5′ TC-3′ for member A3A and 5′-CC-3′ for member 3AG), the underrepresentation of these dinucleotides found in viral genomes has been considered as proof of the long-term effect of APOBEC3 deaminases on viral genomes. According to some estimates, more than 22% of total viral sequences analyzed showed the “footprint” of 3A3 or 3AG activities, suggesting a broad implication of APOBEC3 in shaping the nucleotide composition of viral genomes and in the prevalence of U-ending codons in some viral families [31]. Interestingly, APOBEC3 expression is particularly high in the human respiratory tract (https://www.proteinatlas.org/ (accessed on 23 July 2025)), which can be further induced in primary airway epithelial cells treated in vitro with IFN [30]. This may explain why human respiratory viruses show extremely high U/C rates. Regarding ADAR-1 activity on viral genomes, the A-to-I editing of viral mRNAs may alter decoding by tRNAs since inosine can be paired preferentially with C, followed by U, and, to a lesser extent, with A. Adenine deamination may lead to an A-to-G mutation if ADAR-1 is acting on viral genomic RNA. The role of ADAR-1 in shaping nucleotide composition in some viruses has been confirmed; nevertheless, its effect on virus replication may be detrimental or beneficial depending on the virus analyzed [32].

Another driver of CUB is the general suppression of CpG and UpA dinucleotides found in animal viruses and vertebrate genomes [14,33,34]. For CpG, the depletion initiates with the methylation of cytosine (mC), which may further undergo a deamination to thymidine, thus increasing the frequency of UpG. CpG and UpA represent pathogen-associated molecular patterns (PAMPs) that can trigger the cell’s antiviral response through the action of zinc finger antiviral protein (ZAP) and RNAase L, respectively [35,36]. In general, CpG depletion in viral RNA genomes is more intense than ApU suppression, driving the net effect on synonymous codon usage toward a reduction in C/G-ending codons, as observed in SARS-CoV-2 [18,37]. This viral escape from CpG/UpA-mediated surveillance has made many viruses relatively insensitive to ZAP or RNAse L activities. One exception is the Flavivirus and Togavirus families, which alternate replication in evolutionarily distant hosts (vertebrates and mosquitoes), showing few signs of CpG depletion and CUB in their genomes. Consequently, the replication of these viruses is highly sensitive to ZAP and RNAseL [38,39,40].

Much less is known about the effect of chemical modification of viral RNAs—such as N6-methyladenosine (m6A), 5-methylcytosine (m5C), N4-acetylcytosine (ac4C)—on the translation of viral mRNAs. The m6A is the most prevalent epigenetic modification found in viral RNA genomes, although the effect on virus replication can vary significantly depending on viral species analyzed and the location of the chemical modification in the genome [41,42,43]. Thus, both enhancement or repression of viral translation and RNA stability were reported when the epigenetic profile of viral RNAs was modified, suggesting the existence of a complex functional interaction between virus replication and m6A modification that will require further analysis [43].

Depending on whether the effect of a mutation is beneficial, neutral, or deleterious, the rate at which a mutation is fixed in viral genomes can change. Apart from purifying selection on amino acid sequences, other selective forces acting on viral RNA genomes have been described (Figure 2). Thus, the conservation of RNA structural elements in the coding sequences of viral genomes involved in viral RNA replication (e.g., Cre element in picornavirus) can constrain even synonymous codon changes embedded within these structures [44]. In other cases, codon usage drift towards A/U-ending codons could relax RNA structures to facilitate the transition between replication and translation of viral genomes [23,45]. Another selective force relies on the fact that slow decoding codons afford time for the proper cotranslational folding of multidomain-containing viral polyproteins [46]. Thus, the accumulation of suboptimal codons at specific regions of viral mRNAs corresponding to protein-coding domains has been confirmed in picornaviruses (poliovirus, PV, and Hepatitis A, HAV) and other virus families [46,47]. Selective pressures on synonymous codon composition can also shape the structure of viral quasispecies, determining the position of the genome in sequence space, which can further influence mutational robustness and fitness, as demonstrated using recoded PV or Chikungunya virus (CHIKV) [48,49]. The presence of specific synonymous codons in viral genomes can determine the resulting effect of future mutations in these codons, thus affecting the long-term evolutionary trajectory of viruses.

## 4. Effect of Codon Usage Bias on Viral Translation and Fitness

Viral mRNAs are efficiently translated in infected cells to ensure virus multiplication; however, optimal expression of viral genes in uninfected human cells often requires codon optimization, as seen with the *S* gene of SARS-CoV-2 intended for mRNA vaccines [50]. The role of CUB in viral translation and fitness has been classically interrogated by altering codon usage or codon pair frequencies in coding regions of viral genomes [46]. In most cases, recoding of viral genomes involved deoptimization by introducing codons that are underrepresented in the human genome—a modification that exacerbated the already biased parental codon usage of some viruses, such as human respiratory syncytial virus (HRSV) or Influenza A (IAV). Pioneer recoding experiments were performed in PV, a picornavirus that exhibited a moderate codon bias (tAIg = 0.22, GC3 = 49%) (see Figure 1). The authors drastically replaced parental synonymous codons over the structural coding region of PV with those that are underrepresented in the human genome [51]. Recoding of even short stretches of the structural coding sequence of PV resulted in a dramatic reduction in both translation in cell-free HeLa cell extracts and virus infectivity in human cells [51]. In another report, codon deoptimization affecting only nine types of amino acids of the PV genome also reduced virus fitness without altering viral mRNA translation in rabbit reticulocyte lysates [52]. In the case of human influenza virus (IAV), which shows a significant CUB (tAIg = 0.21, GC3 = 44%) (see Figure 1), codon usage deoptimization of specific RNA genomic segments, including HA (hemagglutinin), NA (neuraminidase), and NS (nonstructural protein), strongly reduced translation and, in some cases, the stability of the corresponding mRNAs [53,54]. In another report, codon usage deoptimization of every genomic RNA segment of seasonal H1N1 virus resulted in an attenuated phenotype in human cells but not in avian cells, a differential effect that highlights the host-specific effect of codon usage alterations [55]. In HIV-1 and HRSV, which exhibit strong codon bias, codon usage modification also resulted in the drastic attenuation of viral replication [56,57]. Notably, codon usage and/or codon pair deoptimization also resulted in a strong attenuation of the resulting recoded virus in animal models; a finding that prompted a new approach to generating live-attenuated vaccines [58,59,60]. Since rare codons are expected to induce ribosome stalling and eventually RNA decay, codon deoptimization of viral genomes could also be reducing viral mRNA stability. However, this possibility has not been analyzed to date. The direct causative effect of codon usage on viral attenuation has been seriously questioned in some cases by the fact that codon usage modification also altered dinucleotide composition in viral genomes, particularly CpG and UpA content [37]. This uncontrolled side effect could be making the recoded virus more vulnerable to host antiviral responses, which likely contributed to the attenuation phenotype observed in vivo. Therefore, a neat demonstration of the effect of CUB on viral mRNA translation in infected cells will require more precise tools to recode viral genomes, perhaps by changing only relevant codons that do not involve altering the CpG or UpA content (e.g., Arg AGA to AGG, Leu CUU/UUA to CUG).

Some differences in RSCUs can also be found across the genome, such that nonstructural genes tend to have less optimal codons than those encoding structural proteins in some Alphaviruses, IAV, and HRSV, among others. In some cases, this correlates with expression levels since viral nonstructural proteins generally accumulate at lower levels than structural proteins. An interesting possibility is that the differential accumulation of viral proteins in infected cells may be regulated, at least in part, by the differential RSCU of the corresponding coding sequences.

## 5. Viral Strategies to Overcome Codon Usage Suboptimality

Despite codon usage suboptimality, viral mRNAs are generally stable and can direct efficient and sustainable translation in infected cells [19]. Since many cytopathic viruses block host translation (the shut-off phenomenon), inducing the degradation of cellular mRNAs in many cases, the remaining translation machinery, including the aminoacyl-tRNA pool, becomes fully available for viral translation. This may counteract codon usage suboptimality as, in the absence of host mRNA competition, even viral mRNAs with low tAI could be efficiently translated by an increase in aminoacyl-tRNA supply (Figure 3). This counteracting effect could explain some previous observations, such as that the translation of viral mRNAs is less efficient outside the context of infection (i.e., compared with transfected cells), or that mutant viruses unable to induce shut-off cannot efficiently outcompete host mRNA for translation [61,62]. A paradigmatic example of this is the differential effect of SARS-CoV-2 NSP1 (Nonstructural protein 1) on translation and stability of host and viral mRNAs. Expression of NSP1 in human cells blocked translation and induced the degradation of host mRNA in a codon usage-dependent manner (Figure 3) [63,64,65]. Thus, mRNAs rich in optimal codons (e.g., actin) were highly sensitive to the inhibitory effect of NSP1, whereas SARS-CoV-2 mRNAs or those cellular mRNAs showing suboptimal codon composition became resistant [63]. This discriminative effect of NSP1 also operated in cultured cells infected with SARS-CoV-2 and, notably, NSP1 deletion resulted in a mutant virus with reduced translation and strong attenuation in animal models [61]. Since many human viruses, including IAV, herpes, bunyavirus, flavivirus, and alphavirus, also induce the degradation of host mRNAs, the existence of a similar counteracting mechanism to improve the translation of viral mRNAs with suboptimal codon usage is an attractive possibility that deserves further investigation [66].

Recent data also revealed alterations in the abundance of isoacceptor tRNAs and modifications in the tRNA epitranscriptome after infection with human viruses, including CHIKV, SARS-CoV-2, and Dengue (DENV). This finding has been associated with translational reprogramming to improve viral translation in infected cells [67,68,69]. tRNA molecules undergo extensive base modifications, particularly in the anticodon loop, that can alter the relative affinity of tRNAs for synonymous codons. Thus, position 34 of the anticodon, responsible for wobble pairing, is subjected to extensive chemical modifications, including deamination and methylation, which can now be analyzed by combining nanopore RNA sequencing and mass spectrometry analysis [70,71]. Among the 20 different modifications documented at position 34, the best characterized are a) A34 deamination to inosine (I), which expands the decoding capability of some tRNAs by now pairing with C/U/A-ending codons; and b) U34 modifications with mcm5 (5-methoxycarbonylmethyl) or mcm5S2 (5-methoxycarbonylmethyl-2-thiouridine), which restricts the pairing of U34 to A-ending codons (e.g., LysAAA; Gln CAA; Glu GAA; Arg AGA) [71,72]. Interestingly, the expression of one of the methylases (KIAA1456) involved in mcm5 modification of the wobble U34 was significantly induced after CHIKV and DENV infection, a finding that correlated with an increase in the levels of mcm5U-tRNAs in infected cells [67]. This finding has been interpreted as a readjustment of the tRNA pool to favor translation of viral mRNAs rich in A-ending codons, such as GAA (Glu), AAA (Lys), CAA (Gln), or AGA (Arg), while repressing those ending in G. A similar increase in mcm5U-tRNAs was recently documented in human cells infected with SARS-CoV-2 [69]. Since many viral genomes are rich in U34-sensitive codons, altering the tRNA epitranscriptome could be a common viral strategy to improve translation. Interestingly, a similar enrichment in U34-sensitive codons was found in genes induced by DNA damage, whose expression was also induced by viral infection, suggesting the possibility that viruses could be exploiting an existing host pathway to improve translation [68,73].

## 6. Codon Usage Bias and Viral Evolution

The question of whether the current CUB of viruses primarily reflects a translational selection or results from long-term host-mediated mutational pressure is a matter of debate. Identifying signatures of codon usage selection in viral genomes is challenging due to the coexistence of compositional, structural, and coding information within the same RNA molecule that can be subjected to accidental covariation. However, the strong correlation between nucleotide/dinucleotide composition and CUB suggests that mutational pressure is probably the most important determinant of the codon bias observed in viral genomes [13,14,74]. Supporting this mutational model is the rampant C-to-U mutagenesis observed in currently evolving SARS-CoV-2 genomes, which is attributed to host-mediated genome editing [29]. According to this model, the pervasive effect of cytidine and adenine deamination, combined with the selective pressure to remove CpG dinucleotides from viral genomes, could be inadvertently increasing A/U-ending codons, thus pushing viral mRNAs away from codon optimality. This drifting, initially harmful to the virus, may evolve to a less disadvantageous situation if the virus has an effective mechanism to outcompete host mRNAs for ribosomes and tRNAs, as described before. Thus, viruses with suboptimal CUB—e.g., human coronaviruses (HCoVs), IAV, and HIV-1—exhibit effective mechanisms to block host translation, whereas others presenting a less biased codon usage—e.g., HCV—are unable to block host translation. Further support for the host-mediated mutational model of CUB is the finding that viruses that alternate replication in distant hosts (mammals/arthropods) tend to exhibit a less biased codon usage than close family members adapted to exclusively infect mammals (Figure 1). Since insects lack the cytidine/adenine deamination-dependent antiviral responses found in mammals, the long-term persistence of these viruses in insects could be protecting them from more pronounced codon usage drift.

Alternatively, the CUB of viral genomes could be the result of a translational selection to achieve an optimal trade-off state between translation elongation, protein folding, and an effective translation-to-transcription switch during virus replication [1,23,75]. According to this model, the selection of suboptimal codon usage would improve viral fitness, a phenomenon that has not been proven to date due to the difficulty in modeling realistic long-term virus evolution in cultured cells. Regarding this central question, a recent report showed that in vitro DENV evolution experiments resulted in some fitness gain associated with a slight codon drift toward A-ending codons, particularly on Arg codons [76]. A similar evolution experiment with recoded viruses showing extreme optimal or suboptimal codon bias could shed light on the causal relationships between CUB and viral fitness.

Another important question is whether CUB reflects adaptation to the host or even virus attenuation in some cases. Interestingly, endemic coronaviruses causing the common cold in humans (229E, OC43, NL63, and HKU1) show even higher U/C ratios at the third position of codons than bat or zoonotic coronaviruses, particularly in the case of HKU1. Perhaps this higher U/C ratio reflects more intense episodes of C-to-U mutagenesis during adaptation to humans [29,77]. According to this model, the increasing U/C ratio observed in circulating SARS-CoV-2 variants compared with the original Wuhan-1 isolate could be interpreted as the irreversible trajectory of SARS-CoV-2 toward becoming the fifth human endemic coronavirus. There is no evidence of virulence attenuation in current SARS-CoV-2 variants compared to the parental virus (Wuhan-1) outside the context of massive vaccination; however, it would be interesting to compare the effect of CUB on translation and fitness between the original Wuhan-1 and evolved SARS-CoV-2 variants in the next decades.

## 7. Concluding Remarks

Despite extensive data analysis, it is still not clear whether codon usage bias of viral genomes primary reflects a translational selection to reach an optimal fitness, or represents a viral adaptation to a mutational pressure imposed by the host. The existence of viral compensatory mechanisms (e.g., host translation shut-off) to improve competitiveness of viral mRNAs with suboptimal codon usage, supports the notion that host-induced mutation of viral genomes is the driver force of codon usage bias, and that viral compensatory mechanisms probably arose as an adaptive process during virus evolution. Although the mutational hypothesis currently gathers more consensus, the possibility that codon bias of viral mRNAs evolved to reach an optimal balance among translation rate, mRNA stability, protein folding or even virus escape to host surveillance, is a possibility that could be explored in the future by combining experimental data with predictive deep learning technologies. The role of codon usage bias in shaping not only viral translation, but also viral RNA stability and turnover necessary for virus replication is a possibility that also deserves further investigation. Finally, it might be useful to use of nucleotide composition and codon usage bias of viral genomes as an estimate of long-term of virus evolution in particular hosts, especially for those human respiratory viruses of zoonotic origin as SARS-CoV-2 and IAV.

## Figures and Tables

**Figure 1 viruses-17-01218-f001:**
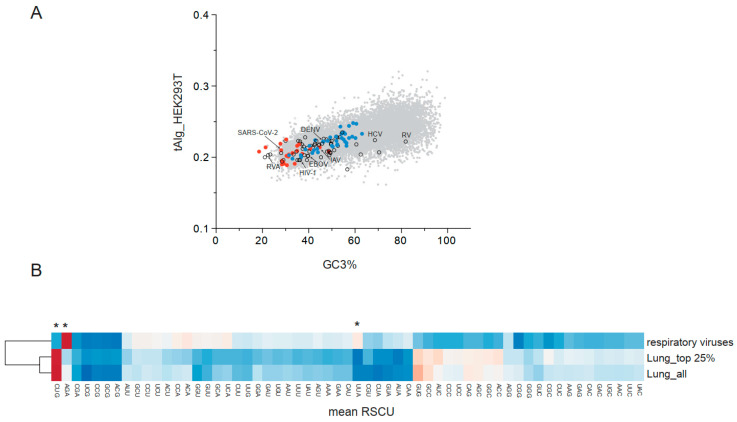
(**A**) Plot showing the transcript-level tRNA adaptation index (tAIg) and the percentage of codons ending in G or C (GC3) in human RNA viruses (colored dots) and in human genes (gray dots). The list of 118 human RNA viruses described in https://viralzone.expasy.org/ (accessed on 23 July 2025) was used. tAIg values are the average of tAI estimated for every codon according to tRNA levels quantified in HEK293T cells [22]. Red dots denote human respiratory viruses, whereas blue dots correspond to arboviruses that infect both humans and insects. Dots corresponding to some viruses that infect humans are labeled; Dengue (DENV), Influenza A virus (IAV), Rubella virus (RV), Hepatitis C virus (HCV), Ebola virus (EBOV), Rotavirus A (RVA), Human immunodeficiency virus (HIV-1), and SARS-CoV-2. (**B**) Mean relative synonymous codon usage (RSCU) value comparison of the twenty-one human RNA viruses causing respiratory illness with all genes expressed in human lung (lung_all), or with the top 25% of expressed genes (lung_top 25%). The heatmap corresponds to Euclidean distance-based clustering according to the color code shown on the right. Main differences were found in the AGA (Arg), CUG (Leu), and UUA (LEU) codons (marked with *).

**Figure 2 viruses-17-01218-f002:**
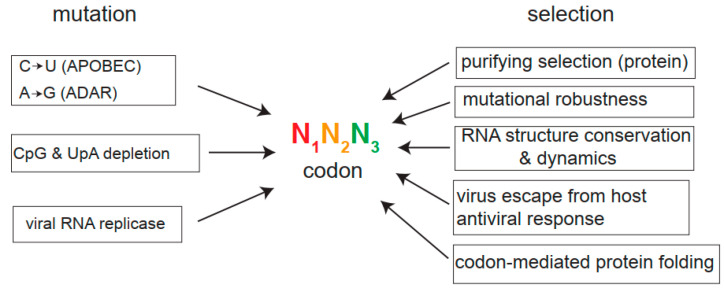
Forces that shape codon usage bias in viruses. A codon is represented as a combination of three nucleotides, the third position (N3, green) being the one that contributes most to synonymous changes as mutations preferably accumulate at N3 during viral evolution. The main mutational drivers, including host deaminases (APOBEC, ADAR), methylation-induced CpG mutagenesis, UpA depletion, and nucleotide misincorporation by viral RNA replicase during replication, are shown. Purifying selection is the main force acting on viral RNA genomes to conserve coding information. Mutational robustness, RNA structure conservation, and dynamics, change to escape the host’s antiviral response, or suboptimal codon-mediated ribosome slowdown to facilitate protein folding are other selective forces described to act on viral genomes.

**Figure 3 viruses-17-01218-f003:**
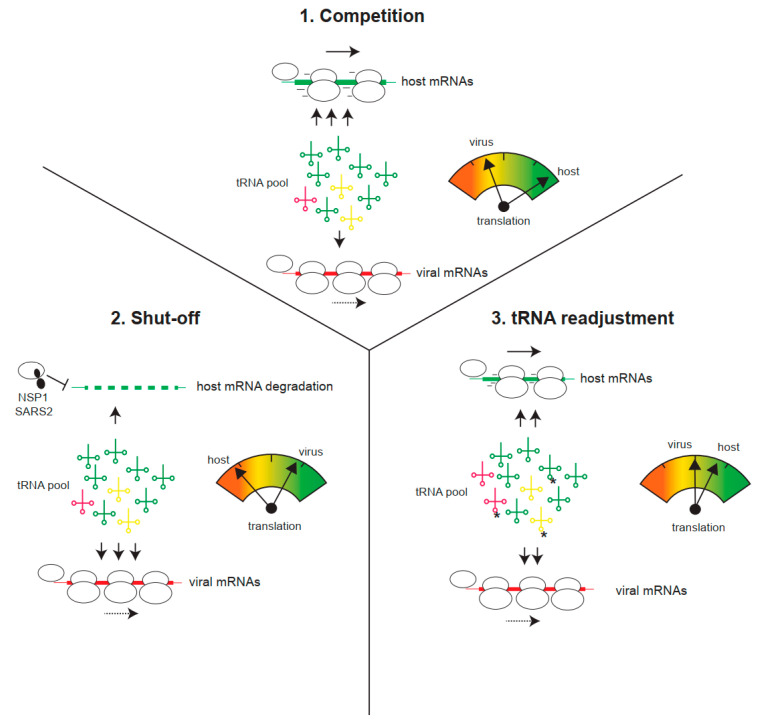
Viral strategies to improve translation of viral mRNAs with suboptimal codon usage. (**1**) Host mRNAs demand the bulk of tRNAs for translation, thus reducing the tRNA pool available for viral translation. Under this competitive situation for tRNAs, viral translation is limited. (**2**) Virus-induced translation shut-off and/or host mRNA degradation (e.g., by SARS-CoV-2 NSP1) reduces the tRNA demand for host translation, thus increasing the availability of tRNAs for viral translation. (**3**) Infection can also induce a tRNA pool readjustment, involving changes in tRNA epitranscriptome (e.g., U34 methylations, labeled as *) or changes in the relative abundance of some isoacceptor tRNAs. This could differentially increase the translation of viral mRNAs.

## Data Availability

Data are contained within the article.

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
