# Peer review of "Codon Usage Bias in Human RNA Viruses and Its Impact on Viral Translation, Fitness, and Evolution"

_viruses, 2025, doi:10.3390/v17091218_

Round 1
Reviewer 1 Report
Comments and Suggestions for Authors
This is an interesting, informative review of a complex literature. While it has many strengths, it could still be improved in some places.
1.Fig. 1B is rather mysterious. What is the color code used in the figure? And what are the lines to the left of the figure, resembling a tree of sequence comparisons?
2.The discussion of the role of Apobecs (lines 110-forward) is confusing. First, “Apobec3A” is usually abbreviated “A3A”, but here I think it repeatedly becomes “3A3”. Second, which Apobec might be involved in editing viral RNAs? The review seems to cite refs. 27 and 28 on this question (line 119), but these references address the role of Apobecs in general and do not implicate specific Apobec family members in their discussions. In contrast, ref. 29 does implicate A3A in SARS-Cov-2 editing. It seems to me that it would be extremely helpful if the review pointed out that A3A has been shown to edit RNA as well as DNA; this is a primary reason (I believe) that A3A is a prime suspect in editing of viral RNA sequences.
3.A minor item: influenza A virus is sometimes “IAV” (the standard abbreviation) but here sometimes AIV as in line 179.
4.Line 185 mentions “a different strategy” for codon deoptimization. This sounds mysterious; perhaps the author would like to devote a sentence or so to a brief description of the strategy.
5.The text contains no reference to Figure 2.
6.Finally, the concluding discussion (line 278) posits “an ecological adaptation to the host” as vs. “long-term host-mediated mutational pressure” as explanations for codon usage bias. I find the phrase “ecological adaptation to the host” totally opaque. I am not sure what the author should do here, as I have no idea what he has in mind, but I think some clarification is needed.
Author Response
1.Fig. 1B is rather mysterious. What is the color code used in the figure? And what are the lines to the left of the figure, resembling a tree of sequence comparisons?.
The reviewer is right, and the missing color code corresponding to distance-based clustering of RSCU is now included. The dendogram shown on the right corresponds to this clustering.
2.The discussion of the role of Apobecs (lines 110-forward) is confusing. First, “Apobec3A” is usually abbreviated “A3A”, but here I think it repeatedly becomes “3A3”. Second, which Apobec might be involved in editing viral RNAs? The review seems to cite refs. 27 and 28 on this question (line 119), but these references address the role of Apobecs in general and do not implicate specific Apobec family members in their discussions. In contrast, ref. 29 does implicate A3A in SARS-Cov-2 editing. It seems to me that it would be extremely helpful if the review pointed out that A3A has been shown to edit RNA as well as DNA; this is a primary reason (I believe) that A3A is a prime suspect in editing of viral RNA sequences.
The reviewer is right and the typo ”3A3” was replaced by A3A. A paragraph on the editing activity of A3A on RNA has been included as suggested by the reviewer. The references are now properly cited.
3.A minor item: influenza A virus is sometimes “IAV” (the standard abbreviation) but here sometimes AIV as in line 179.
The reviewer is right and AIV was replaced by IAV.
4.Line 185 mentions “a different strategy” for codon deoptimization. This sounds mysterious; perhaps the author would like to devote a sentence or so to a brief description of the strategy.
This point has been clarified in the text.
5.The text contains no reference to Figure 2.
Figure 2 is now referenced in the text.
6.Finally, the concluding discussion (line 278) posits “an ecological adaptation to the host” as vs. “long-term host-mediated mutational pressure” as explanations for codon usage bias. I find the phrase “ecological adaptation to the host” totally opaque. I am not sure what the author should do here, as I have no idea what he has in mind, but I think some clarification is needed.
To clarify this point, the phrase “an ecological adaptation to the host” was replaced by “ translational selection” to better expose the two possibilities to explain codon usage bias of viral genomes (selection vs mutation).
Reviewer 2 Report
Comments and Suggestions for Authors
Comments to viruses-3806457
The review manuscript by Ventoso entitled “Codon usage bias in human RNA viruses and its impact on viral translation, fitness and evolution” nicely summarizes the interplay between codon usage bias in RNA viruses and translation, fitness and evolution in their hosts.
The review is timely, easy to follow and addresses an interesting topic that will be of interest for the Viruses readership. In this regard, the manuscript should be accepted for publication once some minor points are addressed.
Minor comments
- The manuscript should include a discussion or concluding remarks section.
- Page 6, lines 150-152: The presence of chemical modifications has been largely demonstrated in viral RNA. Since some of these modifications such as m6A and ac4C have been shown to influence mRNA decoding, translation efficiency and translation-dependent mRNA decay, it would be interesting to add a paragraph on the potential role of RNA modifications on these aspects of viral RNA metabolism.
- Since rare codons are expected to induce ribosome stalling, which instead may trigger deadenylation and RNA decay or ribosome collisions and the ribotoxic stress response, it would be of interest to include some information on the consequences that ribosomal pausing due to codon usage bias has/could have on viral RNA metabolism.
Author Response
Minor comments
1. The manuscript should include a discussion or concluding remarks section.
A concluding remarks section is now included in the text.
2. Page 6, lines 150-152: The presence of chemical modifications has been largely demonstrated in viral RNA. Since some of these modifications such as m6A and ac4C have been shown to influence mRNA decoding, translation efficiency and translation-dependent mRNA decay, it would be interesting to add a paragraph on the potential role of RNA modifications on these aspects of viral RNA metabolism.
Following the reviewer’s recommendation, a short paragraph on this topic has been included in the text (lanes 143-152)
3. Since rare codons are expected to induce ribosome stalling, which instead may trigger deadenylation and RNA decay or ribosome collisions and the ribotoxic stress response, it would be of interest to include some information on the consequences that ribosomal pausing due to codon usage bias has/could have on viral RNA metabolism.
A short paragraph on this topic has been included in the text (lanes 199-201).